# Genome Editing in Human Gametes and Embryos: The Legal Dimension in Europe

**DOI:** 10.3390/biotech12010001

**Published:** 2022-12-23

**Authors:** Takis Vidalis

**Affiliations:** Hellenic National Commission for Bioethics & Technoethics, 10674 Athens, Greece; t.vidalis@bioethics.gr

**Keywords:** genome editing, genetic interventions, Oviedo Convention, human gametes, human embryos, research, reproductive applications, precautionary principle, safety, morality clauses

## Abstract

To date, the legal aspects of the ongoing debate on the application of genome editing in human gametes and early embryos have attracted little attention. In Europe, this seems to have changed with a recent official position that clarifies the meaning of the relevant provision of the common legal instrument on Human Rights and Biomedicine (Oviedo Convention). This provision explicitly prohibits modifications to the genome of future persons and adoptes a precautionary stance with regard to genetic interventions in the human reproductive material. In this article, we examine relevant interpretative options, following the new official clarifications, focusing on the research/clinical application distinctions that characterize their approach. From this viewpoint, we propose an approach that favors basic research activities involving genome editing, even for exploring potential clinical applications under conditions of safety, which may justify a future legislative amendment. Furthermore, we explore the patenting issue, based on the current approach of European case law, and give reasons that may justify patent rights in this ethically sensitive area.

## 1. Introduction

In a recent document [1], the Council of Europe’s Bioethics Committee (CDBIO) revisits the basic legal regulation on genetic interventions in humans as adopted by the European Convention on Human Rights and Biomedicine (Oviedo Convention, 1997) a quarter of a century ago. 

According to the relevant provision (Oviedo Convention, Article 13), genetic interventions for medical reasons are allowed, with the exception of interventions that are intended to modify the genome of future generations. This regulation covered all conventional forms of gene therapy existing at the time of its enactment, although clinical applications were still infrequent, and serious failures occurred as well [2,3] (pp. 164–165). 

In this regulatory landscape, the new genome editing technology represents a real challenge. Genome editing can create targeted breaks in the genome with the use of special enzymes (nucleases), and then can repair these breaks through the function of DNA mechanisms (homologous recombination/HR, and non-homologous end joining/NHEJ). This process makes it possible to intentionally change specific DNA sequences. The new technology involves the use of zinc finger nucleases (ZFNs), TALEN-type nucleases, and CRISPR/Cas9 nucleases. 

Considering its impressive characteristics ensuring easier, very accurate, and much cheaper genetic engineering methodology, nowadays, genome editing is a technological breakthrough. As has already been demonstrated in various fields, from environmental and agricultural interventions [4,5,6] to biomedical applications [7], genome editing has significantly changed the image of modern biotechnology, shaping a future with multiple applications and products. 

Thus far, the astonishing possibilities that this new technology provides towards extensive interventions on the genome of all existing species, including humans, have triggered a wide discussion about the ethical dimension of genome editing [8,9,10]. Questions related to its compliance with generally accepted values (such as the respect for nature, the protection of biodiversity, or even the protection of human dignity) have been raised, and they have raised controversies among scientists, ethicists, and even political decision-makers.

## 2. The Current Approach in European Law

The European Convention on Human Rights and Biomedicine (widely known as the Oviedo Convention) is, currently, the only instrument of international law, with binding force, regulating the field of modern medicine. 

The Oviedo Convention has been adopted by the Council of Europe, an international organization that has developed an advanced legal system, and influences the law in all European countries; the Oviedo Convention is a regional international instrument that produces common regulations for its member states. However, as it happens, with any international legally binding instrument, to produce legal effects at the national level, the Oviedo Convention needs to be signed and then ratified by the parliament of member states wishing to adopt it. 

Conventional medical practice and clinical research are both addressed by the Oviedo Convention’s provisions that form a general regulatory framework detailed by additional protocols pertinent to specific fields. Protocols are also legally binding instruments presupposing that a country has adopted the Oviedo Convention and, following a separate process, it has signed and ratified the protocol of interest. Four such protocols on the prohibition of human cloning, transplantations of human tissues and organs, biomedical research, and genetic data for health purposes have already been enacted, providing more developed regulation in these areas. 

This framework sets the basic legal principles and rules under which we must understand the normative approach of genetic engineering in humans (including genome editing techniques) at the European law level. However, it is not the only one; legal instruments with binding force that regulate the broader field of human rights (particularly the European Convention on Human Rights at the level of the Council of Europe and the European Union’s Charter of Fundamental Rights), as well as important non-binding international instruments of the so-called “soft law” (particularly the two UN’s Universal Declarations on Bioethics and on Human Genome), must also be taken into consideration. 

Still, the Oviedo Convention holds a prevailing position, here, as it includes the only binding provision (that of Article 13, under the title “Interventions on the Human Genome”) specifically dedicated to the regulation of genetic engineering in humans. Article 13 states explicitly:
“*An intervention seeking to modify the human genome may only be undertaken for preventive, diagnostic or therapeutic purposes and only if its aim is not to introduce any modification in the genome of any descendants*”.

Following that formulation, from a legal perspective, gene therapies are, in principle, welcomed as new promising innovations on the condition that their performance has no intention to affect the genetic profile of future persons. Furthermore, given the broader regulatory framework mentioned above, any such intervention must comply with several established legal principles and rules. More specifically, under the framework set by Article 13, genetic interventions are considered to be medical acts of preventive, diagnostic, or therapeutic purpose, the performance of which presupposes the informed consent of the person concerned (Articles 5 and 6 of the Oviedo Convention and Article 3 of the EU Charter of Rights). If such interventions are performed in a clinical research context, they need to comply with the additional requirements set by the Oviedo Convention (Articles 15–17). According to the latter, research is allowed if: (a) no alternative research method exists; (b) potential risks are not disproportionate to expected benefits; (c) scientific and ethical approval by competent bodies is in place; and (d) after fulfilling these terms, specific informed consent in written form has been obtained by the person concerned. However, such consent may be withdrawn at any time without limitation.

Regarding the information of persons undergoing such interventions, in particular, a crucial point here is to clarify that genome modifications produce irreversible effects on the DNA and to provide contingency measures for preventing potential risks. Furthermore, as medical acts, genetic interventions involve the management of genetic and health data of the person concerned, which in turn makes it imperative to provide appropriate protection. In that sense, a number of other provisions are applicable, namely those ensuring medical secrecy (Article 10 of the Oviedo Convention, which is binding for the attending physician) and data protection legal provisions regarding the confidentiality of genetic and health data processing (according to the General Data Protection Regulation, which is in force in the EU, and the relevant national laws). 

At a more general level, these provisions must be aligned with the existing legal principles mentioned in the first articles of the Oviedo Convention. These principles recognize (a) the preeminence of the interests of the human being over the interests of science and society (Article 2), (b) equal treatment in medical care of appropriate quality (Article 3), and (c) the need for any medical intervention to comply with professional obligations and standards (Article 4). In the context of genetic interventions, the application of these principles means, respectively, that:(a)Their performance is unjustified if not expected to benefit the health of the person concerned, even if such performance may improve scientific knowledge in that area,(b)Access to these medical acts must be guaranteed for everyone, and it is ensured that they are of good quality. For that purpose, coverage by social security and the national health system must also be considered.(c)The quality of genetic interventions in terms of safety and efficiency must fulfill evidence-based criteria, and it must be guaranteed through the development of relevant medical protocols setting good clinical practice rules. To the extent that a significant portion of undesired effects of genetic modifications [11,12] still exists (off-target DNA editing), this condition is of major importance.

## 3. The Exception of Interventions on Gametes and Embryos

The Oviedo Convention firmly excludes from any intentional genetic intervention the human reproductive material, that is, reproductive cells and embryos in vitro. 

The rationale for this reservation is grounded on the fact that there is wide scientific uncertainty about the potential effects that genetic modifications on reproductive material may have on the genome of future persons, although successful applications in viable embryos have been reported [13]. On that point, we must stress a generally accepted principle of modern international law in relation to the applications of innovative technologies. 

Originating from environmental law (with regard to the management of genetically modified organisms, in particular), the so-called “precautionary principle” is now a legal standard [14,15] (pp. 141–144) that should be taken into account in medical applications as well [16]. According to this principle, when potential risks of a particular technological application are uncertain, “reasonable measures” must be taken to avoid the potential emergence of these risks (UN Convention on Biological Diversity/CBD, 1991, preamble). The meaning, here, is that it is not necessary to detect risks with scientific certainty; even if this possibility remains scientifically uncertain, this is a sufficient condition for justifying a legal obligation to take preventive measures. Of course, this obligation should not be interpreted in a way that may impede the development of new technology [17] (p. 5). On the contrary, preventive measures also include research intending to overcome uncertainty, that is, to reveal the possibility of concrete risks, based on solid scientific evidence, and thus to establish conditions for safe application [18] (p. 237). 

Furthermore, the term “reasonable” [measures] indicates that, legally, the preventive policy needs to have resulted from a ”balancing interest” exercise with regard to the application conditions. In that view, excessive prohibiting measures may not be legally justified where compelling interests in favor of the new technology exist, such as a significant benefit expected for its user’s health or the public health system, the environment, etc. Likewise, “reasonable” measures must be considered to be those that do not impede the progress of research, that is, do not create the impression of “taboo” areas not to be touched by scientific knowledge.

Given the current state-of-the-art, genome editing techniques continue to raise significant issues of scientific uncertainty, even regarding their applications to non-human organisms or to non-reproductive human cells. It is plausible to assume that such issues are multiplied when it comes to applications to human reproductive material, as research evidence demonstrates so far [19,20]. In this sense, there is room, here, to apply the precautionary principle considering measures towards potential risk avoidance. Such risks are notably off-target damages, and deleterious DSB (double-stranded DNA breaks) repair bioproducts including large deletions, vector integrations, and chromosomal translocations [21], as well as plasmid and retrotransposon insertions [22] when referring to CRISPR-Cas9 applications. Since the first announcements on the scientific potential of the new technology, a similar approach has been adopted in relevant discussions held at various international forums, suggesting a moratorium on genome editing applications in human gametes and embryos [23,24].

This stance does not exclude, in principle, research activities towards overcoming current uncertainties and, eventually, towards justifying the performance of genome editing intending to serve reproductive purposes. However, even though genetic modification of gametes or embryos is freely allowed, research activities are justified insofar as there is a strict condition prohibiting the transfer of modified embryos into the human body for development in vivo, with the aim of reproduction. This is a clear criterion that distinguishes research from potential reproductive applications of that technology. 

Moreover, it is a criterion that falls within the scope of Article 13; indeed, Article 13 tacitly accepts research activities involving human reproductive material since it only prohibits the intentional modification of the genome of “any descendants”, that is, of future persons, after completion of the reproductive process in vivo. Research activities remain outside the scope of this prohibition, even if they “intend” to modify the genome of early forms of human life, since these forms are not considered “persons” (and thus, “descendants”), according to European law as currently in force. It is worth noting, here, that the European Court on Human Rights has explicitly rejected claims defending the recognition of human embryos as subjects of human rights based on that normative ground [25].

Nevertheless, research involving genome editing in embryos (not gametes) must comply with Article 18 of the Oviedo Convention regulating research on embryos in vitro. According to that article:(a)Research on embryos in vitro is allowed on condition of the “adequate protection of the embryo”;(b)The creation of embryos for research purposes is prohibited.

The meaning, here, is that national legislators can allow research only on “spare” embryos initially created but never used for reproductive purposes. Furthermore, the reservation on the embryo’s “adequate protection” seems to exclude, in practice, any experimentation resulting in embryo destruction. A more favoring research interpretation would merely require the informed consent of the gamete donors prior to the use of embryos, namely, the possibility to refuse such use for reasons of embryo protection [26] (p. 63). In any case, the above regulation is not applicable in countries where the Oviedo Convention has not been incorporated into national law (Belgium, UK, Germany, etc.). 

Consequently, some first conclusions regarding the current state of European law regarding the possibility to perform genome editing in human reproductive material are as follows:

For countries adhering to the Oviedo Convention, genome editing may be allowed by national law only in a research context, that is, not for reproductive purposes. In a research context, genome editing in embryos can be performed on two conditions: On the one hand, the embryos are initially created but never used for reproductive purposes, that is, they are not meant to serve research purposes at the onset, and gamete donors have provided advance informed consent for such use. On the other hand, genome editing in gametes can be performed freely, provided that the modified gametes are not meant to be used for the creation of embryos, since Article 18 explicitly prohibits the creation of embryos for research purposes. 

For countries not adhering to the Oviedo Convention, national law is the only regulation that matters and may observe or forsake the conditions mentioned above, both regarding embryos in vitro and gametes. Thus, any research project involving genome editing falls within the prohibition of embryo research in Germany, Austria, Italy, etc., whereas it may be executed, for example, in the UK and Sweden. In the latter countries, the law also allows the creation of embryos for research purposes, which may involve the use of genetically modified gametes.

## 4. Revisiting Oviedo: A New Approach?

The recent CDBIO document reexamines that regulation apparently under the light of the new promising era for gene therapy that genome editing methods (particularly CRISPR-Cas9) represent nowadays. 

According to its first conclusion: Taking into account the technical and scientific aspects of these developments, as well as the ethical issues they raise, it considered that the conditions were not met for a modification of the provisions of Article 13. However, it agreed on the need to provide clarifications, in particular, on the terms “preventive, diagnostic, and therapeutic” and to avoid misinterpretation of the applicability of this provision to “research”.

This preliminary position is complemented by three paragraphs in the meaning of “clarifications”, consisting, in fact, of an official interpretation of Article 13; this interpretation, derived from the expert body, has been presented to the Committee of Ministers of the European legislator (the Council of Europe). It is worth noting here that, from a legal perspective, any law may be subject to various interpretations from the courts, the academic community, the legal experts, or even the citizens. Yet, a “privileged” interpretation is that of legislators themselves (be it a national parliament, an administrative authority, or an international organization such as, for instance, the Council of Europe). If the legislators feel the need to detail their will for justifying or explaining a certain legal provision in order to ensure efficient regulation, they provide an “authentic” interpretation [27]. That interpretation prevails over any other, to the extent that it reflects the genuine will of the legislator. This is exactly the case regarding the CDBIO’s clarifications regarding the meaning of Article 13. In this context, even though the document does not recommend an amendment of the relevant article, it nevertheless leaves room for reflection in terms of its interpretation and application. 

The first of the two clarifications relates to the article’s “scope with regard to research”, which is of interest regarding, in particular, genetic interventions on reproductive material (the other clarification detailing interventions on somatic cells). 

The document brings up two relevant positions. On the one hand, aligned with the argument presented above, it firmly distinguishes research from procreation by rejecting the use of genetically modified gametes, embryos, or their precursors for the purposes of procreation. By contrast, this means that genetic interventions on reproductive material are allowed exclusively in cases where they serve research purposes. Moreover, that position is a step beyond the position of paragraph 91 of the Convention’s Explanatory Report, which accepts interventions in gametes only, not in embryos [28]. However, this means that embryos created originally but never used for reproductive purposes may be subjected to genome editing for research purposes. On the other hand, using gametes submitted to genome editing for creating embryos in a research context remains to be a prohibited option, given the explicit wording of Article 18, paragraph 2.

In addition, the document links that approach to the requirement existing in the above paragraph of the Convention’s Explanatory Report. According to this, the approval of a regulatory or ethical body is required regarding research involving genetic modifications in human gametes. Given that embryos are also included in such research purposes, that approval involves relevant research activities as well. 

However, the most interesting point here remains vague, because the law does not set the criteria upon which relevant decisions of these regulatory or ethical bodies should be based. 

Undoubtedly the first conditions that a research team must fulfill in order to obtain approval are to guarantee that (a) no embryo creation will be performed when genome editing is to be applied to gametes, and (b) in the case of genome editing interventions on “spare” embryos, no implantation for reproductive purposes is planned; both guarantees presuppose monitoring during the research process and until the modified gametes or embryos’ have been destroyed after completion of the experimentation. 

A question is raised about whether a further condition for a project’s approval must be considered, regarding safety issues. In principle, the answer is negative, given that research is the only purpose here; if informed consent has been obtained by the gametes or embryos’ donors, the “fate” of the reproductive material submitted to experimental procedures is of no interest in terms of the law. Yet, a research project in that field is likely to focus on the reproductive material’s safety (and its viability, in particular) if intending to provide evidence that genome editing methods may also serve reproductive purposes in the future. In fact, that prospect would successfully address the requirements of the precautionary principle; the achieved evidence on safety will reduce uncertainty when implementing that technology, making at least visible (and preventable) potential risks that may occur. Moreover, if research can offer evidence on safety, then revisiting the existing regulation and eventually accepting the use of genetically modified reproductive material for procreation also, could reasonably be anticipated. 

The best solution, here, is to establish two levels of approval by the competent regulatory or ethical bodies. At the first level of approval, the only requirement could be appropriate guarantees precluding the creation of embryos from modified gametes and the implantation of modified embryos. At the second level of approval, a requirement on accepted evidence on safety could be considered, only for projects intending to demonstrate that modified gametes or embryos may serve reproductive purposes if allowed in the future. Following this solution, freedom of research (guaranteed by Article 13 of the EU Charter of Rights) would not be circumvented, while there would also still be room for research intending to further explore the potential of genome editing for clinical applications under controlled conditions. 

## 5. The Patenting Issue

Patenting genome editing applications to human reproductive material also raises questions of legal interest.

Within the EU, the most relevant legal instrument is Directive 98/44/EC that regulates patenting in biological applications. Another more general legal instrument (the European Patent Convention) regulates patenting in Europe, albeit making no specific references to biological inventions.

The important feature that Directive 98/44 adopts is the introduction of “morality clauses” as an additional condition for granting patent rights to biological applications, besides the regular, general conditions of patentability (Article 6). According to the European Patent Convention (17th ed., 2020, in Article 52, paragraph 1) “*patents shall be granted for any inventions, in all fields of technology, provided that they are new, involve an inventive step and are susceptible of industrial application*”.

In addition, the Directive’s “morality clauses” require fulfillment of the criteria of “ordre public” and (social) “morality”. This means that a biological product or method may be considered to be unpatentable if these criteria are not fulfilled, even if the regular criteria have been met. The Directive mentions four such cases indicatively, among which are “*processes for modifying the germ line genetic identity of human beings”, and “uses of human embryos for industrial or commercial purposes*”.

This explicit reference means that such processes are excluded from any consideration of patentability by default, namely that no compelling interests could be taken into consideration for justifying patentability whatsoever. 

A question, here, is whether the term “human being” applies (apart from human persons) to gametes and also early embryos. The Directive does not clarify this particular issue, neither explicitly nor in its preamble (no 40). An available reference to other legislative documents is Article 2 of the Oviedo Convention, stating that “*[T]he interests and welfare of the human being shall prevail over the sole interest of society or science*.”. According to recital 19 of the Convention’s Explanatory Report “*[h]uman dignity and the identity of the human being had to be respected as soon as life began.*”. In principle, this particular interpretation is not binding when it comes to the meaning of an EU Directive’s article, given that the EU has not signed the Oviedo Convention. Therefore, the term “human being” remains with no explicit legal definition.

However, the term “human being” clearly does not refer to gametes, since, from a legal perspective, life begins after the moment of conception. Thus, genetically modified gametes may be patentable as such, even if the creation of embryos with their use is excluded. These gametes constitute an innovation and biological invention, in the sense that their modification through genome editing does not exist in the natural world. They also may be subject to industrial exploitation in the context of research procedures exclusively (as any use for reproductive purposes is banned). Finally, they do not fall within the restricted area of morality clauses, given that they cannot be considered “human life” or “human beings”.

Does the same apply to genetically modified human embryos? It seems that we can answer affirmatively in that case also. Certainly, there is no question that an embryo in vitro constitutes a form of “human life” to the extent that, after conception, the process of continuous biological development begins. Yet, a necessary condition for the individualization of an embryo is its implantation in the uterus. Before that moment, it would be logically absurd to consider an embryo a “human being” for two reasons: (a) no differentiation between embryonic and placenta cells exist and (b) it remains unknown whether a split of that embryo after implantation will occur, resulting in more than one individual. For these reasons, even if we consider that the term “human being” pertains to embryos too, this could be valid only at a certain moment of their development, namely after implantation, and never from the very first moment of conception. If we accept that argument, early embryos in vitro that have been subjected to genome editing for research purposes may be patentable too, similar to the patentability of modified gametes, as analyzed above.

Nevertheless, the patentability of human embryos submitted to biological processing has been rejected altogether by a well-known judgment of the European Court of Justice (ECJ), the supreme court of the EU [29]. The judgment focuses not on the Directive’s prohibition of the germ line modification but on the other explicit provision that considers “*uses of human embryos for industrial or commercial purposes*” unpatentable. According to the Court, that prohibition applies to embryos used for research purposes only since patentability refers, by definition, to commercial exploitation. However, the judgment explicitly accepts that “*[o]nly use for therapeutic or diagnostic purposes which is applied to the human embryo and is useful to it [is] patentable*” (paragraph 46). This position is not convincing for two reasons: (a) If by definition, patentability refers to commercial exploitation, this necessarily applies to therapeutic or diagnostic uses too, which makes the judgment’s reasoning inconsistent. (b) If therapeutic and diagnostic purposes can be distinguished from industrial and commercial purposes in a meaningful way in the context of patent law, as the Court argues, there is no reason to refuse the same for research purposes, and therefore, to accept patentability for the latter too.

However, even by following the judgment’s position, we can conclude that embryos that have been submitted to genome editing interventions are subject to patent rights on condition that the research serves therapeutic or diagnostic purposes that may benefit these embryos to the extent of their in vitro life, no matter whether they are meant for reproduction or not. 

## 6. Conclusions

In Europe, the legal regulation of genetic interventions on gametes and embryos is characteristically rigorous. The legal principle of human dignity, existing in most national Constitutions, and the EU Charter of Fundamental Rights (Article 1), along with similar concepts (such as the “primacy of the human being” in Article 2 of the Oviedo Convention) are the guiding normative reasons here. The essence of the human dignity principle rejects any instrumentalization of human individuals, following the famous Kantian axiom [30], (p. 80). Under this light, the current law adopts a precautionary stance, even if it is difficult to consider gametes or embryos in vitro as individuals deserving respect and protection of their human dignity.

Nevertheless, the recent official elaboration on Article 13 of the Oviedo Convention clarifies that research activities must remain free of relevant restrictions provided that no reproductive purpose with the use of this genetically processed material is envisaged. Even if this approach is not associated with a proposal for the article’s amendment, it is, nevertheless, an important regulatory view, particularly because it considers not only gametes but also embryos, as potential subjects of genetic interventions.

The current precautionary attitude needs more scientific evidence to support a future legislative intervention. Such evidence would be reasonably expected only if research is developed under conditions of legal certainty. This is the added value of the official clarification. Hopefully, this could also apply to the patenting issue: Patent rights are the real motives for insisting on research, that is, for bringing closer the prospected scientific evidence on genetic interventions in the early forms of human existence.

## Data Availability

Data sharing not applicable. No new data were created or analyzed in this study. Data sharing is not applicable to this article.

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
