# Peer review of "Genome Editing in Human Gametes and Embryos: The Legal Dimension in Europe"

_biotech, 2022, doi:10.3390/biotech12010001_

Round 1

Reviewer 1 Report

Since the discovery of CRISPR-Cas facilitated the easy modification of target DNA, genome editing has attracted increasing attention as a new approach to treat several refractory diseases. Genome editing therapy for germline cells is currently forbidden due to ethical and safety concerns. In this paper, the author aims to contribute to the ongoing discussion on legal aspects in human gametes and early embryos genome editing by exploring key issues related to the initial applications of CRISPR in reproductive medicine.

Recently, the Steering Committee on Human Rights in Biomedicine and Health (CDBIO) has completed the final step in the process of re-examining Article 13 of the Convention on Human Rights and Biomedicine (Oviedo Convention) by adopting clarifications regarding the scope of the provisions governing research and the purpose limits of any intervention in the human genome. This author provides a timely explanation of the new change in this law. In addition, the author also focus on the issue of CRISPR patents, suggesting that patenting the application of genome editing for human reproductive material also raises issues of legal interest. This paper not only provides a valuable perspective on the legal aspects of advancing gene editing research and applications in Europe, but may also provide a good reference for other countries to establish regulations for gene editing.

Minor revision:

Please change CrisprCas9(line 213 ) to CRISPR-Cas9.

Author Response

Thank you for this favorable review!

Reviewer 2 Report

1.In the introduction, author should include more backgrounds on Genome editing, including TALEN, CRISPR-Cas or so (PMID: 32051598).

2.For the genome editing by CRISPR-CAS9 Technology, the author is suggested to mention the potential clinical side-effect besides the on-target activity, which is absolutely should be considered for law. Specifically, DSBs induced by CRISPR-Cas9 can lead to larger genomic rearrangements including large chromosomal deletions, inversions or translocations (PMID: 34365511), as well as plasmid and retrotransposon insertions (PMID: 32095517 and 35760782)

Author Response

Thank you for your valuable comments.